# Immune Reconstitution Inflammatory Syndrome with Recurrent Paradoxical Cerebellar HIV-Associated Progressive Multifocal Leukoencephalopathy

**DOI:** 10.3390/pathogens10070813

**Published:** 2021-06-28

**Authors:** Paola Frattaroli, Teresa A. Chueng, Obinna Abaribe, Folusakin Ayoade

**Affiliations:** 1Jackson Memorial Hospital, 1611 NW 12th Ave, Miami, FL 33136, USA; paola.frattaroli@jhsmiami.org (P.F.); teresa.chueng@jhsmiami.org (T.A.C.); obinna.abaribe@jhsmiami.org (O.A.); 2Division of Infectious Diseases, University of Miami Miller School of Medicine, Miami, FL 33136, USA

**Keywords:** paradoxical PML-IRIS, cerebellar PML-IRIS, treatment PML-IRIS

## Abstract

Progressive multifocal leukoencephalopathy (PML), presenting as immune reconstitution inflammatory syndrome (IRIS), is a known complication of antiretroviral therapy (ART) in people living with HIV (PLWH). Typically preceded by ART initiation, IRIS may appear simultaneously/unmasked (PML-s-IRIS) or as a delayed/worsening/paradoxical (PML-d-IRIS) presentation of known PML disease. Primary cerebellar tropism continues to be a rare presentation, and paradoxical cerebellar involvement of PML-IRIS syndrome can be a challenge for both diagnosis and management. Steroids have been suggested as a possible therapy in severe cases but the duration of steroid therapy remain elusive. Our case is that of a 34-year-old man with newly diagnosed HIV simultaneously found to have cerebellar PML. His PML lesions however worsened after initiation of ART (PML-d-IRIS) with evidence of increased intracranial pressure. Despite initial favorable response to a short duration of steroids, he had multiple recurrence of his PML lesions after steroids were discontinued. The presence of predominant cerebellar lesions and the question of how long steroids should be provided to prevent or minimize PML recurrence is the highlight of our case. This report emphasizes the need for more controlled studies to assist clinicians in the optimal diagnosis and management of PML-IRIS in PLWH.

## 1. Introduction

Progressive multifocal leukoencephalopathy (PML) is a debilitating complication of infection by John Cunningham virus (JCV) culminating in an often-fatal demyelinating central nervous system (CNS) disease [1]. Incidence among the general population remains 0.22 per 100,000 despite widespread use of antiretroviral therapy (ART) [2]. Notwithstanding the recent epidemiological shift of PML afflicting mostly patients with profound immunosuppression [3], such as those on immunotherapies or monoclonal antibody treatments [4], people living with HIV (PLWH) remain an important subgroup for PML incidence. This is particularly relevant when the CD4+ lymphocyte count is <200 cells/μL [5].

While JCV tropism for the cerebellum is more common among HIV-PML, only few case reports detail the paradoxical type of PML-IRIS (immune reconstitution inflammatory syndrome), particularly those involving the infratentorial lesions which are at higher risk for clinical deterioration [3]. The present case report examines the clinical and radiological progression of IRIS with recurrent paradoxical cerebellar HIV-associated PML and a brief literature review.

## 2. Case Report

A 34-year-old Haitian man with no past medical history presented with initial complaints of ataxia. He reported an unprotected sexual encounter 4 years ago with a suspected HIV-positive partner and was subsequently newly diagnosed with acquired immune deficiency syndrome (AIDS). CD4 cell count was 37.56 cells/mm^3^ [6.83%], CD8 357.82 cells/mm^3^, CD4/CD8 ratio 0.1 and plasma HIV viral load 459,020 copies/mL.

Brain magnetic resonance imaging (MRI) on admission showed increased T2 FLAIR in the brainstem and cerebellum, as well as bilateral, but asymmetrical involvement of the cerebellar peduncles, not consistent with intracranial mass or acute stroke (Figure 1). Lumbar puncture (LP) showed: white blood cells 2/μL, red blood cells 140/μL, glucose 51 mg/dL, total protein 48 mg/dL, lymphocytes 94/μL, monocytes 6/μL. The cerebrospinal fluid (CSF) polymerase chain reaction (PCR) for JC virus returned as 6400 copies/mL, suggestive of PML. Other CSF studies were unremarkable, including: *Cryptococcus neoformans* antigen, venereal disease research laboratory test (VDRL), *Toxoplasma gondii* PCR, meningoencephalitis panel, *Mycobacterium tuberculosis* PCR, Epstein Barr virus (EBV) PCR, cytomegalovirus (CMV) PCR, herpes simplex virus (HSV)-1 and -2 (PCR), varicella zoster virus (VZV) PCR, adenosine deaminase (ADA), angiotensin converting enzyme (ACE) and acid-fast bacilli culture and smear. Serum serological testing was positive for *Toxoplasma gondii*, but otherwise negative. Other negative blood tests include EBV PCR, CMV PCR, hepatitis panel, and interferon gamma release assay. CSF cytology and flow cytometry were negative for malignancy including B- or T-cell lymphoma.

High concern for PML was raised; less likely toxoplasmosis or IRIS, as the patient was not on ART at the time. Differential considerations for his imaging appearance included CNS lymphoma, neurosarcoidosis, and cerebritis-meningitis among others. The patient was initiated on bictegravir 50 mg/emtricitabine 200 mg/tenofovir alafenamide 25 mg and prophylaxis against opportunistic infections notably Pneumocystis carinii pneumonia. After 2 weeks of ART, the viral load decreased significantly to 1400 copies/mL.

Two months after the initial presentation and despite compliance to ART, the patient was re-admitted for progressive neurological complaints, including worsening ataxia, now unable to stand or walk without assistance. The physical exam was notable for right facial droop, decreased sensation over the right face, marked dysmetria with finger-to-nose testing, and dysdiadochokinesia. He had significant ataxic gait but full muscle strength and negative pronator drift or nuchal rigidity. Repeat CSF studies were unrevealing. Brain MRI was concerning for worsening of previous PML lesions with multifocal brainstem and cerebellar T2/FLAIR hyperintense lesions, new extensive leptomeningeal and perivascular enhancement, and mass effect (Figure 2). Repeat T- lymphocyte subsets/HIV studies at this time showed: CD4 cell count 252 cells/mm^3^ [7.94%], CD8 2430.4 cells/mm^3^, CD4/CD8 ratio 0.1 and plasma HIV viral load decreased to 106 copies/mL. In the setting of radiologic and clinical progression of disease with recently diagnosed AIDS and ART initiation 1 month prior, paradoxical cerebellar PML-IRIS was diagnosed. While ART was continued, he was also started on an adjunctive intravenous methylprednisolone course 1 g daily for 5 days, followed by an oral prednisone taper 60 mg per day, with taper over 6 weeks and close outpatient follow up. The patient was discharged after 5 days of hospital stay.

Patient did not return to receive care at our institution, but had 2 readmissions at an outside hospital within 20 miles to our hospital facility. The first readmission after discharge from our institution occurred within one week with symptoms of nausea, vomiting and gait dysfunction. Brain MRI again showed right cerebellar vasogenic edema with enhancing lesions in the left cerebral peduncle and extension to the pons, left cerebellar peduncle and right medullary edema. Repeat LP studies at this time was positive for JC virus PCR. He was treated again with a course of intravenous corticosteroids. Follow up CD4 count was 101 cells/mm^3^ and sulfamethoxazole/trimethoprim was continued for opportunistic infection prophylaxis. The patient was subsequently discharged to a rehabilitation treatment center after 4 weeks hospital stay.

One month after leaving the rehabilitation program, the patient was readmitted again with symptoms of nausea, vomiting and blurry vision. Repeat brain MRI showed a new 6 mm enhancing focus in the medulla in addition to imaging findings very similar to his last hospital stay. Work up for lymphoma and other malignancy was negative even though CSF studies were positive for EBV PCR in addition to JC virus. Patient again got a course of intravenous steroids. A repeat MRI performed after completion of intravenous steroids showed significant improvement of lesions. He was discharged home with outpatient physical therapy and his functional status continues to improve as of the time of this report.

## 3. Discussion

The incidence of PLWH experiencing PML is virtually unchanged despite the pre-ART era [6], with HIV-PML portending a higher mortality compared to PML related to malignancy or autoimmune disease [7]. While ART has both considerably improved mortality in PLWH and served as immune restoration for PML, a portion of known HIV-PML patients develop neurologic deterioration in the setting of IRIS [7]. One systematic review of PML-IRIS in 13,103 PLWH cited a mortality as high as 28% [7].

The concept of *unmasking* and *paradoxical* IRIS has been well described, mostly in association with mycobacterium tuberculosis (TB) after the initiation of ART (TB-IRIS) [8]. These are two manifestations of the same disease: the first is the discovery of a previously undiagnosed/subclinical disease after starting ART (unmasking) while the second is the worsening clinical and/or radiographic features of a prior or known condition after the initiation of ART (paradoxical) [9].

Of the dichotomy of PML as it relates to IRIS, the first involves the simultaneous unmasking of both PML and IRIS (PML-s-IRIS) as new diagnoses in patients who were previously asymptomatic [3]. The second is termed delayed PML-d-IRIS indicating pre-existing PML that paradoxically worsens after introduction to ART despite a significant recovery in immunologic response and virologic evidence of improved HIV suppression [3]. It is unclear whether dysregulated immune reconstitution contributes to the worsening process of PML or if JCV infection progresses despite immune recovery [3].

This case report elucidates the trajectory of HIV-associated IRIS with paradoxical worsening of PML. A unique observation in our case is the recurrent nature of the cerebellar lesions (at least thrice) compatible with PML-IRIS after initial satisfactory response to steroids. This raises the question whether high-dose steroids should be provided in PML-IRIS for a more prolonged period before tapering to reduce the risk of recurrence. This should be a focus for future retrospective or prospective studies.

Consistent with the literature that 50–73% of HIV-PML patients have predominantly motor deficits [7], our patient experienced worsening ataxia with marked dysmetria, and new onset cranial neuropathy concurrently with diffuse leptomeningeal and perivascular enhancement throughout the cerebellum and brainstem. Cerebellar lysis of oligodendrocytes and astrocytes with subsequent demyelination [10] results in the most severe presentation of HIV-PML as incoordination often precludes independent living [7]. Parallel to a retrospective review of 54 PML-IRIS cases, our patient presented with progression of his syndrome between 1 week and 26 months after exposure to ART [3].

Another important observation in our case which is also encountered by many HIV clinicians is the difficulty in diagnosing PML-d-IRIS (PML delayed IRIS) [5]. While analysis of our patient’s initial CSF revealed JCV DNA, the repeat lumbar puncture upon his worsening presentation during his admission to our institution did not detect JCV (even though follow up CSF studies at the outside facility detected JCV PCR). Exposure to ART will progressively decrease the viral load and improve patient’s immunosuppression. This improvement in immunosuppression may have contributed to the negative LP for JC virus during the first readmission of our patient, further increasing the challenge of making the correct diagnosis. Our case also illustrates the need for follow up CSF studies if the initial analysis is non-diagnostic. 

Diagnosis of PML-IRIS among PLWH can be achieved by invasive brain biopsy with histopathologic demonstration or a combination of clinical, radiologic, and virologic evidence. The latter may include CSF positivity for JCV DNA with an increase in CD4 count and decrease of HIV viral load [5]. In particular, PML-d-IRIS is suggested 2 weeks to 4 months [11] after ART initiation with contrast-enhanced lesions on brain magnetic resonance imaging (MRI) [9]. Since our patient fits into this latter picture, brain biopsy was not pursued as all the major elements to make a diagnosis of PML-d-IRIS were present. High suspicion for the possibility of PML-d-IRIS should be entertained in the setting of down-trending HIV viral load, T-cell restoration, unfavorable radiologic and clinical progression, with negative workup for other infectious etiology [7].

In a retrospective analysis of PLWH, Tan and colleagues noted that compared with the subgroup of PML-s-IRIS, those with PML-d-IRIS had earlier detection of IRIS. This could indicate the priming of their immune system to pre-existing antigens of JCV warranting a shorter time to IRIS presentation [3]. PML-d-IRIS also correlated radiographically with higher lesion loads, decreased survival time and worse mortality [3]. The presence of contrast enhancement and mass effect, such as seen in our patient, is suggestive of PML-IRIS, rather than classic PML, as it alludes to an inflammatory compromise of the blood–brain barrier [3]. Typical manifestations of PML involve the frontal and parieto-occipital regions, though concomitant posterior fossa structures have been described [7]. Even more rare are lesions exclusively involving the cerebellum and/or brainstem as in our patient [9]. Much of the current data on cerebellar lesions describe unmasking, rather than paradoxical PML-IRIS among PLWH [12]. Notwithstanding, D’Amico and colleagues reported a case of PML-d-IRIS whose dysarthria and ambulation paradoxically worsened before eventual stabilization with time [13]. Mossakowski et al. described an autopsy series of 20 HIV seropositive patients, four of whom had exclusively posterior fossa involvement, suggesting cerebellar PML may constitute a separate clinical entity different from cerebral PML and that strains of JCV exist with predisposition to infect specific brain regions [9]. While not seen in our case, granule cell neuronopathy shares a similar predisposition of exclusive infection of the cerebellum, typically causing localized atrophy among PLWH and CNS manifestations without white matter lesions [9]. 

There is a paucity of literature describing the syndrome of PML-d-IRIS and, subsequently, no evidence-based guidelines exist for prevention or management. The use of steroids to treat PML and its optimal duration and dose are also unclear [3], although it is established that early and high dose corticosteroid use over weeks duration should generally be reserved for life-threatening cases of diffuse inflammation, imminent brain herniation, and severe neurological deficits [11]. In addition, our case illustrates that multiple courses of intravenous steroids may be needed for optimal management of PML-IRIS and prolonged high dose steroids may reduce the risk of recurrence.

Since no targeted treatment or prophylaxis exists against JCV, bolstering of the immune system with ART constitutes a primary approach to treatment of classic PML [10]. However, life-threatening PML-d-IRIS may require high-dose glucocorticoid therapy [7] with a treatment strategy that targets the inflammatory response [11].

In conclusion, providers and clinicians who treat HIV should be familiar with the different manifestations of PML in the setting of IRIS and the need for prompt intervention including corticosteroids which may be life-saving.

## Figures and Tables

**Figure 1 pathogens-10-00813-f001:**
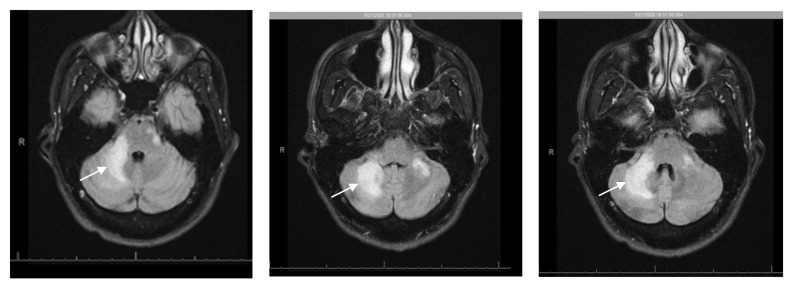
Initial magnetic resonance imaging (MRI) brain demonstrating high T2, and high FLAIR signal in the brainstem (midbrain and pons), inferior cerebellum and bilateral (right more than left) middle cerebellar peduncles suggestive of a demyelinating process (arrow).

**Figure 2 pathogens-10-00813-f002:**
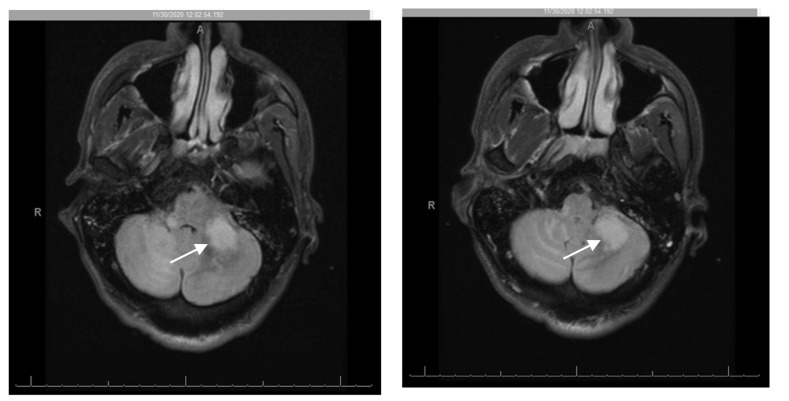
Repeat MRI brain 2 months after compliance with antiretroviral therapy (ART) showing interval worsening of multifocal brainstem and cerebellar T2/FLAIR hyperintense lesions with mass effect. New diffuse leptomeningeal and perivascular enhancement in the cerebellum (arrow), brachium ponti, pons, and left midbrain.

## Data Availability

The data presented in this study are available on request from the corresponding author.

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
