# Peer review of "Immune Reconstitution Inflammatory Syndrome with Recurrent Paradoxical Cerebellar HIV-Associated Progressive Multifocal Leukoencephalopathy"

_pathogens, 2021, doi:10.3390/pathogens10070813_

Round 1

Reviewer 1 Report

The authors present a case of recurrent PML-d-IRIS and its management. The case report is well discussed in relation to existing literature and treatment standards.

Major comments:

  1. At what point was the patient discharged from the authors’ institution? Also, do the authors know if the patient complied with their oral steroid tapering regimen?
  2. Are dosage details of intravenous corticosteroids administered to the patient during the two outside admissions available? The authors should provide these details.
  3. During the two outside admissions, was the patient prescribed oral steroids upon being discharged?

Minor comments:

  1. Line 34, microL is not a standard abbreviation and should be replaced with uL.
  2. Line 49, please add units.
  3. Line 178, predated is not the correct word here and should be replaced with pre-existing.
  4. The manuscript needs language editing, especially for faulty grammar.

Author Response

Thank you for your detailed reviewed. Below you would find a response to each point:

Major comments:

  1. Supply the date of discharge - Dec 4, 2020. Patient was discharged after 5 days of hospital stay.
  2. We do not have information about details regarding oral steroid tapering from the outside facility. Since he went to a skilled setting, medication compliance is typically supervised.
  3. We can confirm that he got IV dexamethasone but we cannot confirm the dose he received from the outside records. 
  4. We cannot confirm the patient got oral steroids upon being discharged from the outside facilities but we can confirm he completed a course of IV steroids at our facility and the two outside admissions.

Minor comments:

  1. Line 34, microL is not a standard abbreviation and should be replaced with uL: Done
  2. Line 49, please add units: Done
  3. Line 178, predated is not the correct word here and should be replaced with pre-existing: Done
  4. The manuscript needs language editing, especially for faulty grammar: reviewed.

Reviewer 2 Report

This manuscript reports a PML-d-IRIS case and suggests the optimal treatment strategy for patients living with HIV.

  1. CSF should be abbreviated in line 50 instead of line 51. 
  2. How the percentage [%] of "CD4 cell count" (lines 45 and 76) is calculated?  37.56 is 6.83%, then 252 is 11%?
  3. The sentence in line 83 could be improved.  "Patient was lost to follow up at our institution," doesn't make sense.
  4. Line 92: Change to "...after a 4-week hospital stay"
  5. Line 97: Replace "was" with "were", since it's referring to CSF studies.
  6. Figures could be formatted to have the same way, some show the date on top, others do not. Also, they can be aligned with a uniform. In Figure 1, the first figure does not look like T1. The authors need to check again if this is the correct T1 scan.

Author Response

Thank you for your detailed reviewed. Below you would find a response to each point:

  1. CSF should be abbreviated in line 50 instead of line 51: Done
  2. How the percentage [%] of "CD4 cell count" (lines 45 and 76) is calculated?  37.56 is 6.83%, then 252 is 11%? 
    1. Line 45: CD4 cell count was 37.56 cells/mm3 [6.83%], CD8 357.82 cells/mm3, CD4/CD8 0.1
    2. Line 76: CD4 cell count was 252 cells/mm3 [7.94%], CD8 2430.4 cells/mm3, CD4/CD8 0.1  
    3. Correction made regarding CD4% from 11% to 7.94%. Added CD8 abs count and CD4/CD8 ratio.
  3. The sentence in line 83 could be improved.  "Patient was lost to follow up at our institution," doesn't make sense: Patient did not return to received care in our institution.
  4. Line 92: Change to "...after a 4-week hospital stay": Ok
  5. Line 97: Replace "was" with "were", since it's referring to CSF studies: Done
  6. Figures could be formatted to have the same way, some show the date on top, others do not. Also, they can be aligned with a uniform. In Figure 1, the first figure does not look like T1. The authors need to check again if this is the correct T1 scan: Corrected. Figures refers to T2, and high FLAIR.